# Identification and Characterization of the *StCPAI* Gene Family in Potato

**DOI:** 10.3390/plants14162472

**Published:** 2025-08-09

**Authors:** Zhiqi Wang, Wenbo Wu, Tao Liu, Wenting Shi, Kai Ma, Zhouwen He, Lixuan Chen, Chong Du, Chaonan Wang, Zhongmin Yang

**Affiliations:** 1College of Horticulture, Xinjiang Agricultural University, Urumqi 830052, China; wangzhiqi1026@163.com (Z.W.); wwb173522984@163.com (W.W.); 13565545234@163.com (T.L.); shiwenting3800@126.com (W.S.); mkaswx@163.com (K.M.); 17699098818@163.com (Z.H.); c1346433748@163.com (L.C.); godv2018@163.com (C.D.); wcn0107@126.com (C.W.); 2Postdoctoral Station of Horticulture, Xinjiang Agricultural University, Urumqi 830052, China

**Keywords:** potato, *Carboxypeptidase A inhibitor* genes, auxin response

## Abstract

Carboxypeptidase A inhibitor (CPAI) is a globular polypeptide that specifically inhibits carboxypeptidase A activity in the insect gut, playing a vital role in plant defense against external stimuli. To date, this gene family has not been systematically characterized in potatoes. In this study, we identified the *CPAI* gene family using the potato DM v6.1 genome and analyzed genomic and amino acid sequence features. Results demonstrated that eight *CPAI* members in potatoes share high homology with orthologs in tomatoes, eggplants, and peppers. Their promoter regions contain predicted cis-acting elements associated with defense and stress responses. Additionally, qRT-PCR analysis revealed elevated expression of specific members in tubers and aerial tubers, with concurrent responses to auxin treatment. These findings provide a foundation for elucidating the roles of *StCPAI* genes in potato development.

## 1. Introduction

Potatoes (*Solanum tuberosum* L.), an annual dicotyledonous plant in the genus Solanum of the family Solanaceae, ranks as the world’s third-largest staple food crop. Alongside wheat, maize, and rice, it constitutes the four major staple crops globally, playing a pivotal role in alleviating regional food shortages, reducing poverty, and enhancing food security [1]. Its tubers are nutritionally dense, rich in starch, protein, dietary fiber, vitamins, and minerals. It also serves as an industrial raw material for papermaking, cosmetics, adhesives, textiles, and plastic production, and is processed into animal feed, alcohol, and biofuels [2].

Plants employ diverse defense mechanisms to combat pathogens. Structural barriers, such as the cuticle surrounding cells and lignin deposition within cell walls, function as physical and chemical barriers against pathogen invasion [3,4]. Among them, the pathogenesis-related (PR) proteins constitute essential components of the plant defense system. Key members include thionins (PR-13), defensins (PR-12), lipid transfer proteins (LTPs, PR-14), and protease inhibitors (PR-6), all critically involved in suppressing phytopathogens [5,6,7].

As a subclass of protease inhibitors (PR-6), carboxypeptidase inhibitors (CPIs) are globular polypeptides purified from potatoes. They specifically inhibit carboxypeptidase activity in the insect gut and are induced by abscisic acid (ABA) and jasmonic acid (JA) treatments, with prominent accumulation in tubers and wounded leaves of potato plants [8,9]. Insect digestive systems primarily contain two classes of carboxypeptidases: carboxypeptidase A (CPA) and carboxypeptidase B (CPB). Carboxypeptidase inhibitors (CPIs) specifically suppress CPA activity within the insect midgut [10,11].

Studies have also demonstrated that carboxypeptidase inhibitors (CPIs) exhibit antifungal activity against fungal pathogens in vitro. When expressed in transgenic rice, CPIs enhance resistance to fungal infections [12]. Research on carboxypeptidase inhibitors (CPIs) is more extensive in tomatoes (*Solanum lycopersicum*). The *CPI* homolog *TCMP-1* (*Solyc07g007250*), a metallocarboxypeptidase inhibitor, encodes a 37-amino acid polypeptide. It exhibits high expression in floral buds, with transcriptional upregulation in leaves upon wounding [13]. In tomatoes, the metallocarboxypeptidase inhibitor gene *Solyc01g067295.1.1* possesses a promoter that mediates tissue-specific expression in type-VI glandular trichomes [14]. Evidence also suggests that carboxypeptidase inhibitor (CPI) genes potentially modulate floral development in tomatoes [15]. *Solyc07g007250.4.1*, an isoform of *TCMP-1* in tomatoes, is implicated in modulating cadmium accumulation and orchestrating antioxidant responses in plants [16]. A similar response occurs in tobacco (*Nicotiana tabacum* L.), where metallocarboxypeptidase inhibitors *NtMCPIa* and *NtMCPIb* respond to cadmium (Cd) stress by exhibiting differential upregulation in Cd-treated leaves [17].

In this study, we identified the *StCPAI* gene family from the potato DM v6.1 genome. Through integrated bioinformatic analysis and quantitative real-time PCR (qRT-PCR), we characterized this carboxypeptidase A inhibitor gene family and quantified its expression profiles. Six *StCPAI* genes exhibited distinct spatiotemporal expression patterns across various organs and in response to hormone treatments. Moreover, the prominent accumulation of *CPAIs* in potato tubers [8,9]—the plant’s primary economic organ responsible for nutrient storage and yield—suggests potential specific roles beyond general defense, possibly in tuber development, physiology, or resilience to tuber-specific stresses. Therefore, systematic characterization of the *StCPAI* family is crucial to uncover these potato-centric functions, laying a foundation for deciphering the biological functions of the *StCPAI* family in potatoes.

## 2. Results

### 2.1. Identification and Chromosomal Distribution of the StCPAI Gene Family

Gene encoding carboxypeptidase A inhibitor was retrieved from the Spud DB Potato Genomics Resource by querying the conserved domain, Pfam ID PF02977. Through Hidden Markov Model (HMM) searches, we identified eight genes annotated as carboxypeptidase A inhibitor proteins. These were named *StCPAI1* to *StCPAI8* based on their physical order on chromosomes (Figure 1). Chromosomal mapping revealed distribution on chromosomes 1, 3, and 7, with six members physically clustered at both termini of chromosome 7. Among these, *StCPAI3*, *StCPAI4*, *StCPAI5*, *StCPAI7*, and *StCPAI8* displayed tandem duplication (marked by red connecting lines), suggesting this mechanism primarily drove *StCPAI* family expansion.

### 2.2. Physicochemical Properties, Subcellular Localization, and Protein Secondary/Tertiary Structure Prediction of StCPAIs

The predicted physicochemical properties (Table 1) of StCPAI proteins showed amino acid residues ranging from 80 to 125, molecular mass spanning 9.03 kDa (StCPAI5) to 13.93 kDa (StCPAI1), pI values between 5.32 (StCPAI4) and 8.58 (StCPAI7), instability indices from 21.65 (StCPAI2) to 58.52 (StCPAI7), aliphatic indices of 70.00 (StCPAI4) to 94.48 (StCPAI6), and grand averages of hydropathicity (GRAVY) between −0.362 (StCPAI6) and −0.004 (StCPAI4), collectively indicating that most StCPAIs are unstable alkaline hydrophobic proteins (instability index > 40 denotes unstable proteins) [18]. Subcellular localization predictions placed all members in the extracellular matrix, while secondary structure analysis revealed no β-turns in any member. Tertiary structure predictions via homology modeling using SWISS-MODEL (Figure 2) achieved >80% sequence identity between templates and targets for all eight proteins.

### 2.3. Phylogenetic Analysis of StCPAI

To clarify the evolutionary relationships of *StCPAI* genes, a phylogenetic tree was constructed for 26 CPIA domain protein sequences from solanaceous crops: tomatoes (SlCPAI, 10), peppers (CaCPAI, 3), and eggplants (SmCPAI, 5), using HMM-based searches and NCBI–CDD conservation filtering (Figure 3). These proteins were grouped into four clusters (I–IV). In cluster III, SlCPAI2 (with StCPAI1) is specifically expressed in tomato trichomes, and its promoter is linked to type VI glandular trichomes in cultivated tomatoes [14,19]. Alleles of SlCPAI3 (with StCPAI2) have a significant genetic effect on lycopene content and may importantly influence lycopene biosynthesis [20]. In cluster IV, SlCPAI7 (with StCPAI4) is significantly up-regulated in the unilateral incompatibility (UI) reaction, while SlCPAI9 is down-regulated; they contribute to reproductive isolation by affecting pollen tube growth or interactions with pistil tissues [15]. SlCPAI8 (with StCPAI5) is a variant of the tomato metallocarboxypeptidase inhibitor I (*TCMP-1*) gene and interacts with HIPP26 to regulate cadmium accumulation and antioxidant responses in plants, playing an important role in response to heavy metal toxicity and stress [16]. The adjacent SlCPAI9 is up-regulated when tomatoes are infested with Tuta absoluta and may be involved in inhibiting pest digestive enzymes and reducing damage to plant tissues, thus playing an important role in plant defense mechanisms [21]. In cluster I, SlCPAI10 (with StCPAI6, StCPAI7, and StCPAI8) is annotated as a fruit-specific protein, and its absence or down-regulation in the tomato R182 subline may affect fruit development [22]. These findings indicate that the *CPAI* gene family is widely involved in plant developmental processes.

### 2.4. Collinearity Analysis Between StCPAI and Tomato SlCPAI Gene Families

In order to clarify the evolutionary relationship between *StCPAI* gene family members, TBtools was used to construct a comparative collinearity map of the potato *StCPAI* family and tomato *SlCPAI* family (Figure 4). The results indicated that there are four pairs of direct homologous *CPAI* genes in the two genomes, located on the same chromosomes: namely, *StCPAI1* and *SlCPAI2* on chromosome 1; *StCPAI2* and *SlCPAI3* on chromosome 3; and *StCPAI3* and *SlCPAI7*—as well as *StCPAI6* and *SlCPAI10*—on chromosome 7. This suggests that the *CPAI* gene families in tomatoes and potatoes have a similar evolutionary relationship.

### 2.5. Conserved Motifs and Gene Structure Analysis of StCPAI Family Members

The gene structure and conserved motif analysis of *StCPAI* members was conducted using TBtools (Figure 5). The analysis revealed that *StCPAIs* exhibit variations in both the number and types of motifs present. *StCPAI1* and *StCPAI2* have Motif1 and Motif2; *StCPAI6*–*StCPAI8* have Motif1-4; *StCPAI3*–*StCPAI5* have Motif1, Motif3, and Motif6; and all members have Motif1, suggesting strong conservation of Motif1. Additionally, all members possess the CarbpepA_inh Conserved Domain, and a signaling peptide is located at the C-terminus of *StCPAI1*–*5*.

### 2.6. Analysis of Cis-Acting Elements in Promoters of StCPAI Gene Family Members

Analysis of cis-acting elements within the promoter regions of *StCPAI* gene family members revealed that all members possess cis-acting regulatory elements involved in light response, ABA (abscisic acid) response, and anaerobic induction, as well as auxin and methyl jasmonate (MeJA) response elements (Figure 6). The number of light-responsive elements is the highest in the promoter regions of all genes. Light-responsive elements mediate the signal transduction of photoperiod, which is crucial for tuber formation in potatoes. The key gene *SP6A*, which regulates tuber formation, is regulated by photoperiod signals [23]. In addition, individual members harbored gibberellin and low-temperature response elements, cis-acting regulatory elements involved in zein metabolism regulation, and salicylic acid response elements within their promoters. Furthermore, the promoter regions of *StCPAI1* and *StCPAI2* contained cis-acting elements associated with defense and stress responses, while *StCPAI6* and *StCPAI7* possessed wound-responsive elements and MYB-binding sites involved in drought induction. This is consistent with the known function of *CPAI* in pathogen defense [10,11,12,13]. These findings together suggest that *StCPAI* family members likely participate in regulating plant growth processes by responding to hormones and external stimuli. Notably, cis-acting elements implicated in endosperm expression and circadian rhythm control were identified in the promoter regions of *StCPAI4* and *StCPAI5*. Collectively, these results indicate that the *StCPAI* gene family members are broadly involved in various plant developmental processes.

### 2.7. Analysis of the StCPAI Protein Interaction Network

To understand the function of the *StCPAI* genes in potatoes, the interactions among the eight StCPAI proteins were predicted using the STRING online database (Figure 7). The results revealed that StCPAI5 and StCPAI8 interact with other proteins. Specifically, A0A0B4J3K8, which interacts with StCPAI8, is annotated as GDP-mannose 3′,5′-epimerase. StCPAI5 was found to interact with ST1, SN2, and PIN2K. Furthermore, the remaining proteins in the network also interact with StCPAI proteins. However, due to a lack of functional annotation, their specific roles have not yet been identified.

### 2.8. Expression Pattern Analysis of StCPAI Gene Family Members in Different Organs and Under Auxin Treatment

To understand the expression of *StCPAI* gene family members in different organs, six members from the family were selected for RT-qPCR, which was used to analyze their expression in leaves, stems, stolons, tubers, and aerial tubers (Figure 8). Results showed that *StCPAI4*, *StCPAI5*, *StCPAI7*, and *StCPAI8* had significantly higher expression in aerial tubers than in stems. *StCPAI2* had the highest expression in stems, followed by aerial tubers, with lower levels in other organs. *StCPAI3* showed higher expression in tubers and leaves, while *StCPAI4* had the highest expression in tubers and leaves, as well as high expression in aerial tubers. *SP6A* expression was also measured and found to be higher in leaves than in tubers, contrasting with *StCPAI3*, which had higher expression in tubers than in leaves.

To investigate the response of *StCPAI* genes to hormone treatment, 10 μmol·L^−1^ auxin was applied to *E172* seedlings in the greenhouse, followed by RT-qPCR (Figure 9). Analysis revealed that *StCPAI2* expression in apical buds and young leaves peaked at 6 h. *StCPAI3* expression in young leaves gradually increased at 6 and 24 h. *StCPAI4* expression in young leaves showed a significant increase after 24 h of IAA treatment. *StCPAI5* expression in apical buds sharply rose at 6 h, with a decline at 24 h, but was still significantly different from the 0-h level. *StCPAI7* and *StCPAI8* had minimal expression changes across organs and time points, though *StCPAI8* expression in lateral leaves slightly increased at 24 h. All genes except *StCPAI7* showed significant differences in expression patterns in apical buds and young leaves across different time points.

## 3. Discussion

This study is the first to identify an 8-member *StCPAI* gene family in potatoes. The members of this family directly participate in the regulation of tuber development through significant organ-specific expression, particularly dominant accumulation in underground and aerial tubers and responses to exogenous auxin. Bioinformatics analysis revealed its conserved structural characteristics, phylogenetic relationships, and enrichment of cis-acting elements in the promoter region in response to various hormones (such as ABA, IAA, MeJA) and abiotic stresses.

All members of the *StCPAI* family encode unstable basic hydrophobic proteins with a conserved single intron structure and core motif, Motif1. Their promoters are generally enriched with light, ABA, IAA, and MeJA response elements, suggesting that the function of this family is conserved in *Solanaceae* plants [17]. The collinearity results indicated that tandem replication phenomena existed in *StCPAI3*, *StCPAI4*, *StCPAI5*, *StCPAI7*, and *StCPAI8*, while the collinearity between species revealed that the *CPAI* genes in potatoes and tomatoes were located on the same chromosome. The phylogenetic tree shows that the *CPAI* genes of potatoes and tomatoes are clustered in similar evolutionary branches, indicating that the *StCPAI* and tomato *CPAI* genes are direct homologous genes and functionally conserved in *Solanaceae*.

Protein–protein interaction prediction indicates that StCPAI5 and StCPAI8 may cooperatively regulate abiotic stress responses and tuber germination by interacting with stress-responsive proteins. The functions of ST1, SN2, and PIN2K interacting with StCPAI5 have been identified. In *Arabidopsis thaliana*, the 14 different *ST1* gene family members encode sulfate transporters, which are expressed in specific membranes and have tissue and developmental stage specificity [24]. Snakin-2 (SN2) has antimicrobial activity and plays different roles in the crosstalk promoted by various biotic and abiotic stress sources, as well as plant hormones [25].

In potatoes, Snakin-2 is an antimicrobial peptide that positively regulates plant disease resistance and is induced by wounding, pathogen infection, and abscisic acid [26]. Overexpression of *StSN2* inhibits tuber sprouting, while the sprouting process is accelerated in *StSN2* RNAi lines. *PIN2K* (Proteinase inhibitor type-2 K) belongs to the potato type-II proteinase inhibitor family, which plays an important role in plant resistance to insects, bacteria, and pathogenic fungi, and is part of the potato defense mechanism [27]. Pin2 inhibits the activity of serine proteases, thereby disrupting the digestive processes of insects and the invasive capabilities of microorganisms, thus protecting plants from harm. In addition to these functions, Pin2 may also be involved in plant responses to drought, programmed cell death, plant growth, trichome density, and branching [11]. *StCPAI5* may coordinate the regulatory process of tuber germination through the SN2/PIN2K complex. The SN2 binding may enable StCPAI5 to inhibit the proteolytic activation of factors promoting sprouting (such as gibberellin biosynthesis enzymes) during wound response [26]. The interaction of PIN2K may facilitate suppression of the cell wall remodeling driven by serine proteases, which is necessary for bud germination. A0A0B4J3K8, which interacts with StCPAI8, is GDP-mannose 3′,5′-epimerase. Zhang found that overexpression of the *GM35E* gene in tomatoes enhanced tolerance to abiotic stress [28]. Similarly, overexpression of *GM35E* in *Arabidopsis thaliana* and *Oryza sativa* also enhanced their tolerance to acid, drought, and salt stress [29,30]. StCPAI8 likely interacts with genes such as *GM35E* and participates in the response of potatoes to abiotic stress.

The expression profiles of the different organs reveal that the members of *StCPAI* have differentiation functions: *StCPAI3/4/5/7/8* are co-expressed in tubers, possibly jointly regulating the development of both underground and aerial tubers. *StCPAI2* has relatively high expression in stems and is also expressed in other parts, such as leaves. This result is similar to the expression pattern of *SlCPAI2* in tomato trichomes, which is in the same evolutionary branch [14]. The expression trend of *StCPAI3* is completely opposite to that of *SP6A*. It is speculated that *StCPAI3* may be expressed in tubers, but its protein is accumulated in leaves. *StCPAI4* may have a function similar to *SP6A*, being expressed in leaves and then promoting tuber development. However, this promoting effect may not distinguish between aboveground and underground tubers, as the expression levels of *StCPAI4* are high in both aerial tubers and underground tubers, while *SP6A* is highly expressed only in leaves and underground tubers. The expression levels of *StCPAI5*, *StCPAI7*, and *StCPAI8* are low in leaves but high in underground and aerial tubers. Combined with their clustered positions on the chromosome and similar structures mentioned earlier, it is speculated that they may collaborate to regulate the development of tubers in both the aboveground and underground parts (Figure 10).

Tubers, as the modified organs of stems, are crucial for potatoes to adapt to the environment, survive, and reproduce. It is worth noting that multiple *StCPAI* members are preferentially expressed in tubers (including aerial tubers). Aerial tubers are a special form of tuber, and they occur in different parts. Underground tubers are formed by the expansion of the tips of underground stolons, while aerial tubers are formed by the expansion of the axillary bud meristematic tissue above ground. Aerial tubers compete with underground tubers for photosynthetic products, resulting in a decrease in the yield of underground tubers [31]. The formation of aerial tubers involves the fine regulation of various hormones, environmental factors, and gene expression. Studies have shown that the key activator of tuber formation, the FT homolog of floricin, *SP6A*, is mainly expressed in leaves induced by short-day conditions, and its expression level in leaves is relatively low under long-day conditions. Meanwhile, the expression of the *SP6A* gene in stolons has a self-regulatory circuit. It is expressed not only in leaves but also in stolons during tuber formation [23]. In this experiment, we found that the expression level of this gene was high in tubers but low in stolons. This might be because the stolons were not fully expanded, and the tubers were not fully formed at the time of sampling.

Future studies should prioritize selecting members of the *StCPAI* gene family that exhibit high expression in tubers while concurrently participating in potato resistance to pests and diseases. Particularly in the China Xinjiang, which is affected by the Colorado potato beetle (*Leptinotarsa decemlineata*), in-depth investigation into the functions of these genes during tuber development and pest resistance processes—with special emphasis on their potential interactive relationships with key tuberization genes *SP6A* and *IT1*—will provide novel insights into deciphering the mechanisms of tuber formation and enhancing potato resistance to pests and diseases.

## 4. Materials and Methods

### 4.1. Plant Materials

The diploid potato line *PG6359*, which exhibits natural self-compatibility, served as the plant material. Single-node stem segments with one attached leaf were cultured in a culture vessel containing Murashige and Skoog (MS) medium supplemented with 2% (*w*/*v*) sucrose and 0.8% (*w*/*v*) agar. Plants were maintained in a greenhouse at 21 °C under a 16 h light/8 h dark photoperiod. After approximately two months, uniformly grown seedlings were selected for tissue sampling. Stems, leaves, tubers, and aerial tubers were harvested. Three biological replicates (each consisting of pooled tissues from five plants) were collected per organ. The sampled materials were immediately frozen in liquid nitrogen and stored at −80 °C for subsequent total RNA extraction and cDNA synthesis, which served as templates for tissue-specific qRT-PCR analysis.

For hormone treatment, the IAA used was indole-3-acetic acid, purchased from PhytoTech Labs (CAS number 87-51-4), and initially dissolved in a minimal volume of anhydrous ethanol (final ethanol concentration in stock: <1% *v*/*v*) to facilitate dissolution. This stock solution was diluted with distilled water to a concentration of 0.01 mol·L^−1^. Before application, the stock was further diluted 1000-fold in distilled water to achieve a final working concentration of 10 μmol·L^−1^ IAA.

The plant material used for the hormone treatment experiment was the diploid potato genotype, *E172* (a hybrid of *S. tuberosum* group *Tuberosum* and *Solanum chacoense*). *E172* seedlings grown in tissue culture vessels were transplanted to seedling trays. After the plants reached approximately 10 cm in height and completed acclimatization, they were transferred to the greenhouse for further growth. Plants were treated when they reached a suitable developmental stage. Treatment group were sprayed 10 μmol·L^−1^ IAA solution and the control group was sprayed with an equal volume of 0.01% (*v*/*v*) anhydrous ethanol solution. Each treatment (including control) was applied to five individual plants.The treatment was applied once. Immediately after spraying, the plants were covered with a transparentplastic film tent to maintain high humidity and minimize hormone volatilization. A mature leaf, apical bud, leaflet, and stem were collected at 0 h(immediately post-covering), 6 h, and 24 h after treatment application. The collected samples were immediately flash-frozen in liquid nitrogen and stored at −80 °C.

### 4.2. Identification of StCPAI Gene Family Members

The *StCPAI* gene family members were identified using the following procedure: the genome sequence, annotation files, coding sequences (CDS), and protein sequences of the doubled haploid potato line DM v6.1 were downloaded from the Potato Genomics Database (SpudDB; https://spuddb.uga.edu/, accessed on 10 June 2025). The Hidden Markov Model (HMM) profile for the PF02977 (carboxypeptidase A inhibitor) domain was downloaded from InterPro (https://www.ebi.ac.uk/interpro/, accessed on 10 June 2025). Candidate genes were initially identified using the HMM search function in TBtools v2.0.41 [32]. Subsequently, protein sequences of candidate members were submitted to NCBI’s Conserved Domain Database (https://www.ncbi.nlm.nih.gov/Structure/cdd/cdd.shtml, accessed on 10 June 2025) for validation to ensure all contained the conserved carboxypeptidase A inhibitor domain. Following this screening, eight genes were ultimately retained as family members.

### 4.3. Chromosomal Localization and Tandem Repeats of StCPAI Gene

In TBtools, click BLAST, select Blast Compare Two Seqs [Sets] <Big File>, add the protein sequence, and align the potato protein sequence against itself. Then, click Synteny Visualization Text Merger for MCScanx, drag and drop the annotation file of the potato genome, and click Start. Afterward, open “Amazing Gene Location” from the GFF3/GTF File, drag in the files generated from the previous steps along with the chromosome IDs, and visualize the results.

### 4.4. Analysis of Physicochemical Properties, Subcellular Localization Prediction, Secondary and Tertiary Structures of StCPAI Protein

The physicochemical properties of the StCPAI protein—including amino acid length, molecular weight, theoretical isoelectric point (pI), instability index, aliphatic index, and grand average of hydropathicity (GRAVY)—were analyzed using the ProtParam tool (https://web.expasy.org/protparam/, accessed on 10 June 2025). Subcellular localization was predicted using WoLF PSORT (https://wolfpsort.hgc.jp/, accessed on 10 June 2025). For both analyses, the FASTA-formatted protein sequence was submitted to the respective online platforms. The secondary structure prediction was performed using the NPSA server (https://npsa.lyon.inserm.fr/cgi-bin/npsa_automat.pl?page=/NPSA/npsa_sopma.html, accessed on 10 June 2025) with the SOPMA method. The tertiary structure was modeled via homology modeling using the SWISS-MODEL database (https://swissmodel.expasy.org/, accessed on 10 June 2025).

### 4.5. Phylogenetic Tree Analysis of StCPAI

For phylogenetic analysis of the *StCPAI* genes, genomic sequences and annotation files for tomatoes, peppers, and eggplants were downloaded from Ensembl Plants (https://plants.ensembl.org, accessed on 10 June 2025). *CPAI* homologous genes in these species were identified using the same method applied to potatoes. Protein sequences were aligned with MUSCLE in MEGA X v11.0.13 software, and a neighbor-joining (NJ) phylogenetic tree was constructed with 1000 bootstrap replicates. The final tree was visualized and annotated using Evolview v2 (https://www.evolgenius.info/evolview-v2/, accessed on 10 June 2025).

### 4.6. Interspecies Synteny Analysis of StCPAI Family Members

Open TBtools, select “Others”, “Plugin”, “Plugin Store”, and click “Install Selected Plugin” for “One Step MCScanX [Super Fast] by CJ”. Use this plugin to import the tomato and potato genome files and generate a synteny diagram.

### 4.7. Conserved Motif and Gene Structure Analysis of the StCPAI Family

Conserved motifs were identified using the MEME suite v5.5.0 (https://meme-suite.org/tools/meme, accessed on 10 June 2025) with default parameters, while gene structure diagrams depicting exon–intron organization were generated via TBtools software. For protein domain prediction, sequences were submitted to both the NCBI Conserved Domain Database (https://www.ncbi.nlm.nih.gov/Structure/cdd/cdd.shtml, accessed on 10 June 2025) and the Pfam database (http://pfam-legacy.xfam.org/, accessed on 10 June 2025), and the resulting domain architectures were integrated and visualized as structural maps using TBtools.

### 4.8. Cis-Acting Element Analysis in StCPAI Promoters

Upstream 2000 bp sequences of all potato coding sequences (CDS) were extracted from the genome using the “GTF/GFF3 Sequence Extract” function in TBtools. Promoter regions for *StCPAI* genes were isolated with the “Quick Fasta Extractor or Filter” module. Sequences were converted to uppercase format via “Fasta Sequence Manipulator”. These processed promoter sequences were submitted to the PlantCARE database (http://bioinformatics.psb.ugent.be/webtools/plantcare/html/, accessed on 10 June 2025) for cis-acting element prediction. The resultant data were collated in Microsoft Excel, then imported into TBtools for visualization of cis-regulatory elements using the “Simple BioSequence Viewer” on the menu.

### 4.9. StCPAI Protein Interaction Network Analysis

Predict protein interactions via the STRING web portal (https://string-db.org/, accessed on 10 June 2025), and visualize the interaction network using Cytoscape v3.9 [33].

### 4.10. Quantitative Real-Time PCR (qRT-PCR)

Total RNA was extracted from all the samples using the RNAprep Pure Plant Kit (DP441, Tiangen Biotech, Beijing, China), followed by cDNA synthesis with the PrimeScript RT Reagent Kit with gDNA Eraser (Takara Bio, China) to eliminate genomic DNA contamination. The RT-qPCR reaction mixture (20 μL total volume) comprised 10 μL TB Green Premix Ex Taq II (Takara Bio, China), 6.4 μL nuclease-free ddH_2_O, 0.8 μL each of gene-specific forward and reverse primers, and 2 μL cDNA template. Amplifications were performed on a Roche LightCycler 96 System (Roche Diagnostics, Basel, Switzerland) with the thermal profile: initial denaturation at 95 °C for 30 s, 40 cycles of denaturation at 95 °C for 5 s, and combined annealing/extension at 60 °C for 30 s. *Actin11* served as the reference gene for normalization, with relative expression levels (tissue-specific values relative to stems) calculated using the 2^−ΔΔCT^ method from three biological replicates. Gene-specific primers targeting *StCPAI* family members were designed based on sequences (Table 2).

## 5. Conclusions

The carboxypeptidase A inhibitor gene family in potatoes was identified and named *StCPAI* based on the conserved domains it contains. Eight members of the *CPAI* gene family in potatoes were analyzed. They were distributed on chromosomes 1, 3, and 7. The proteins were all unstable basic hydrophobic proteins. The phylogenetic tree divided them into four subfamilies, which were directly homologous to the *CPAI* gene in tomatoes. The structures of each member within the family are similar, and the promoter regions have similar cis-acting elements. It was found through RT-qPCR that their expression levels varied in various organs. Some members had relatively high expression levels in tubers and aerial tubers, which might play a role in the process of tuber development. Most members responded to the regulation of auxin.

During the growth of potatoes, tubers, as important sink organs, play a role in storing nutrients during the growth process. For potato plants that underground tubers, the formation of aerial tubers not only consumes more energy but may also affect the normal formation of underground tubers. Previous reports on the *CPAI* gene have mostly focused on insect resistance and disease resistance. Our discovery provides a new perspective for studying the diversity of tuber development and also offers a potential target for future improvement of potato plant architecture.

## Figures and Tables

**Figure 1 plants-14-02472-f001:**
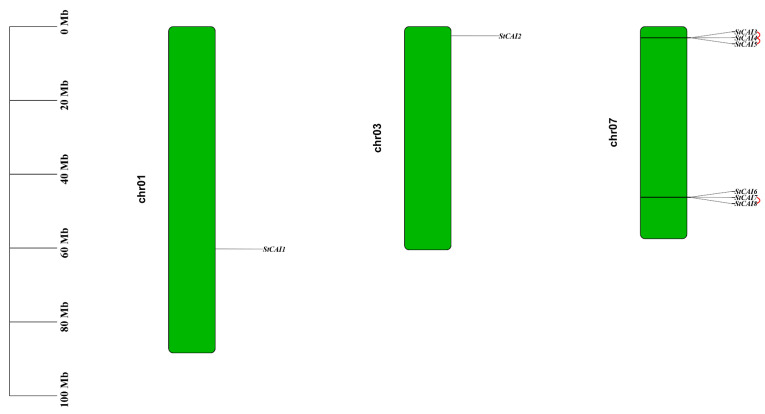
Chromosomal distribution of *StCPAI* genes. Chromosomes are numbered on the left. Gene positions are labeled on the right side of each chromosome. The scale bar on the left indicates physical distance (Mb). Red connecting lines denote tandem duplication events among *StCPAI* genes.

**Figure 2 plants-14-02472-f002:**
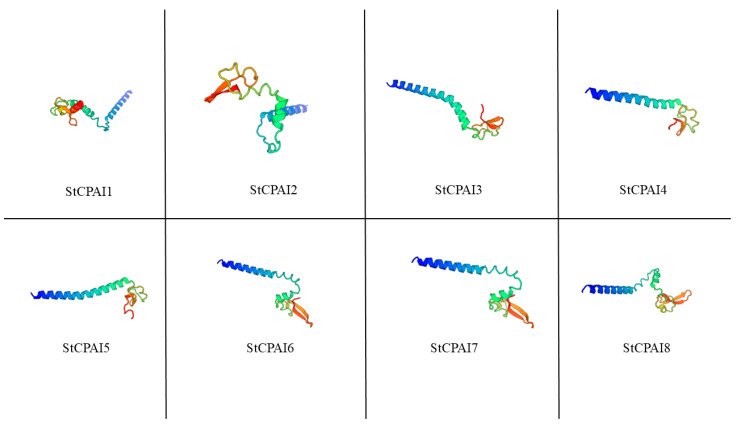
Predicted tertiary structure prediction of StCPAI proteins. Structural models generated by homology modeling using SWISS-MODEL. Proteins are arranged according to their name. Row 1: StCPAI1 to StCPAI4, row 2: StCPAI5 to StCPAI8.

**Figure 3 plants-14-02472-f003:**
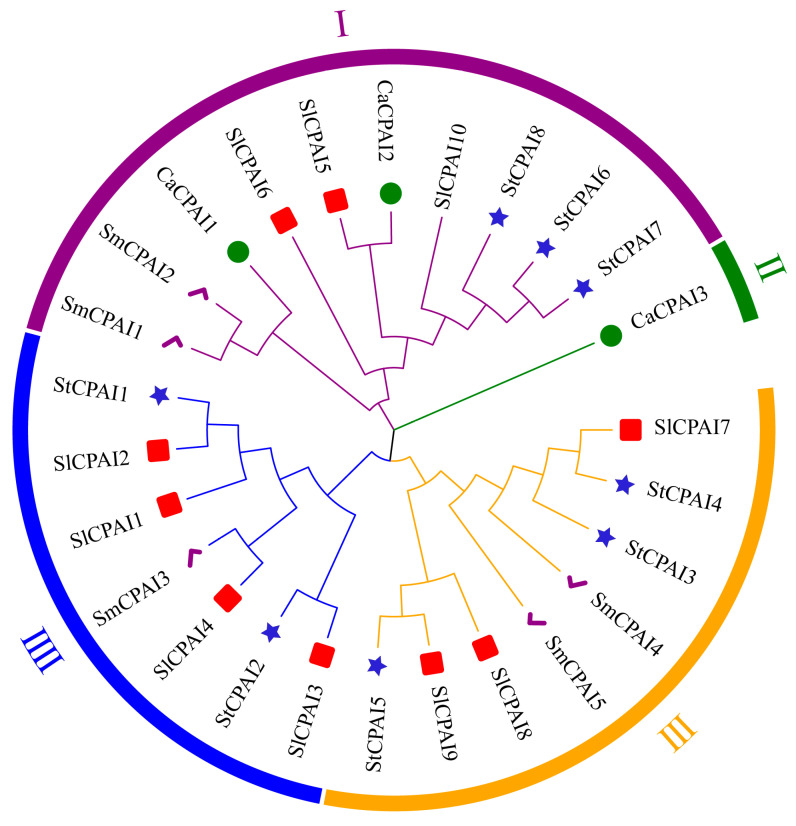
Neighbor-joining phylogenetic tree of potatoes (*Solanum tuberosum*, St), tomatoes (*Solanum lycopersicum*, Sl), eggplants (*Solanum melongena*, Sm), and peppers (*Capsicum annuum*, Ca). The outermost circle colors denote different subgroups. The inner blue stars, red squares, green circles, and purple checks represent potato, tomato, pepper, and eggplant CPAI proteins, respectively.

**Figure 4 plants-14-02472-f004:**
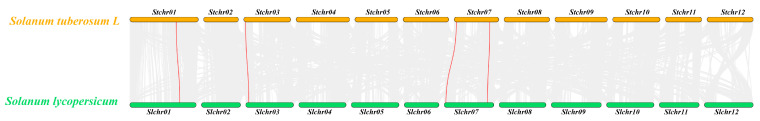
Gene family homology analysis between *CPAI* genes in potatoes and tomatoes; the red line represents genes with homology between the two gene families, while the gray line represents genes without homology.

**Figure 5 plants-14-02472-f005:**
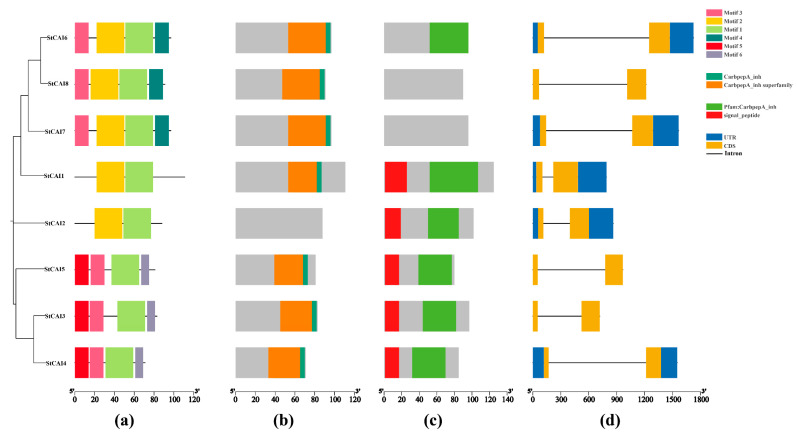
Gene structure and conserved motifs of the *StCPAI* gene family: (**a**) conserved motifs within this family; different colors represent distinct motifs; (**b**) conserved structural domains possessed by family members in NCBI; different colors indicate various conserved structural domains; (**c**) conserved structural domains possessed by family members in Pfam; different colors denote different conserved structural domains; (**d**) gene structures of *StCPAIs*; blue boxes represent un-translated regions (UTRs), and yellow boxes represent coding sequences (CDSs). Lines represent introns.

**Figure 6 plants-14-02472-f006:**
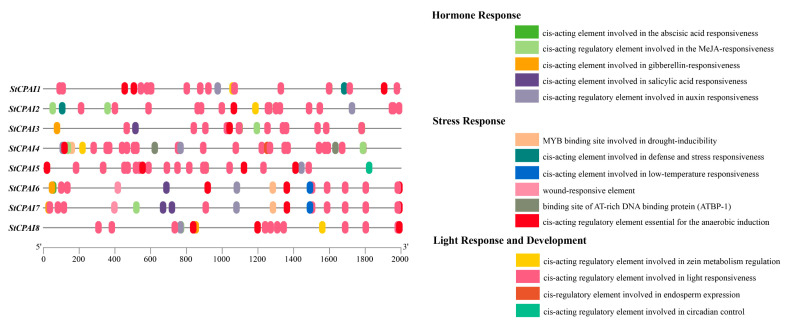
Predicted cis-acting elements in the 2000-bp upstream promoter sequences of *StCPAI* genes. Colored boxes represent various types of cis-acting elements.

**Figure 7 plants-14-02472-f007:**
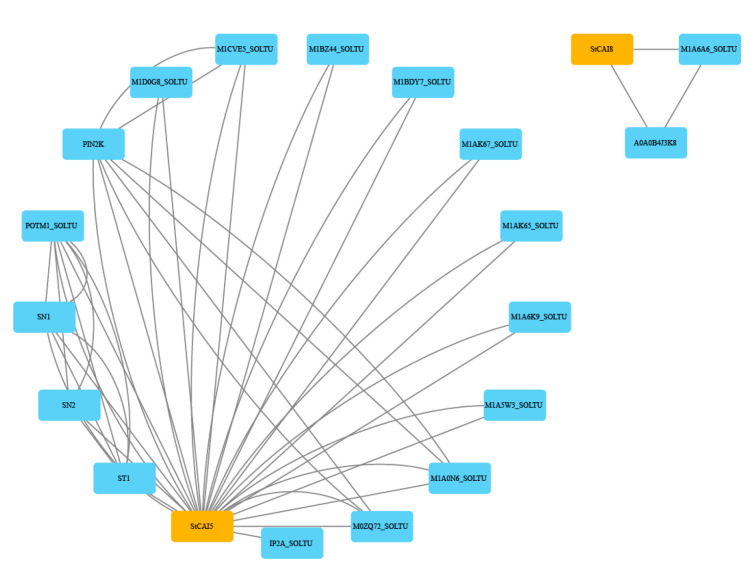
Possible interaction network of StCPAI proteins with other potato genes. Grey lines show computationally predicted protein-protein interactions (PPIs) from the STRING database, with minimum interaction confidence set at 0.4. Nodes represent StCPAI members and their interacting proteins.

**Figure 8 plants-14-02472-f008:**
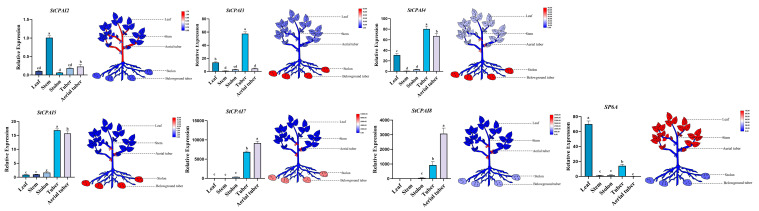
Expression level of *StCPAI* genes detected by RT-qPCR in different organs. Different letters above bars indicate significant differences (*p* < 0.05).

**Figure 9 plants-14-02472-f009:**
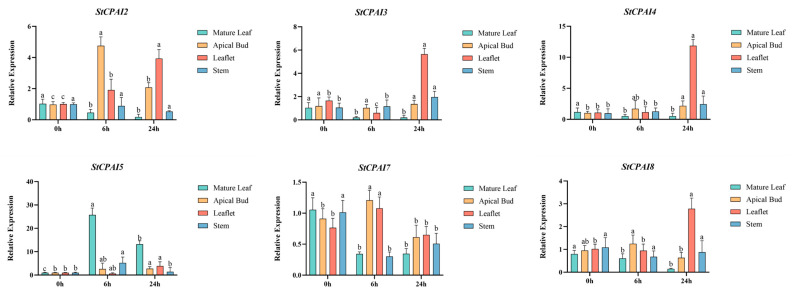
Expression profiles of *StCPAI* genes in response to indole-3-acetic acid (IAA) treatment. Different letters above bars indicate significant differences (*p* < 0.05).

**Figure 10 plants-14-02472-f010:**
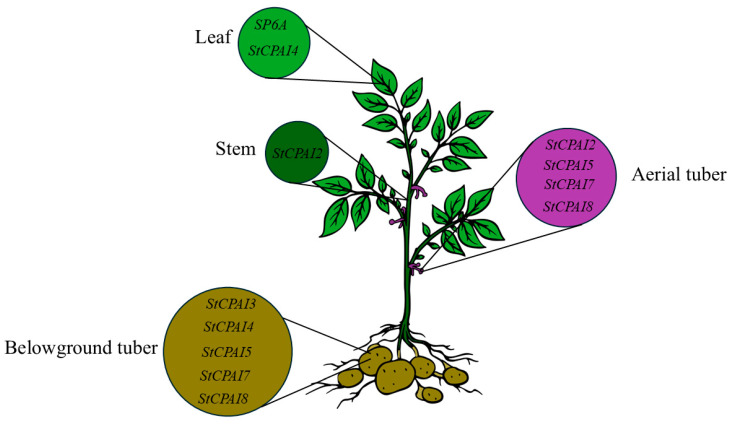
*StCPAI* gene family members and possible functional sites.

**Table 1 plants-14-02472-t001:** Physicochemical properties, subcellular localization prediction, and secondary structure of StCPAI proteins.

Gene ID	Transcript ID	Molecular Mass (kDa)	Acid Amino Length	Theoretical pI	Instability Index	Aliphatic Index	Grand Average of Hydropathicity	Subcellular Localization	Protein Secondary Structure		
									Alpha Helix	Beta Turn	Random Coil
*StCPAI1*	Soltu.DM.01G022210.1	13.93	125aa	5.65	50.04	73.36	0.021	Extracellular	38	0	54
*StCPAI2*	Soltu.DM.03G002590.1	11.5	102aa	7.59	21.65	83.14	0.215	Extracellular	37	0	42
*StCPAI3*	Soltu.DM.07G002650.1	11.24	97aa	5.38	41.39	73.4	−0.105	Extracellular	31	0	39
*StCPAI4*	Soltu.DM.07G002660.1	9.68	85aa	5.32	41.83	70	0.004	Extracellular	30	0	32
*StCPAI5*	Soltu.DM.07G002670.1	9.03	80aa	7.64	43.51	74.5	−0.116	Extracellular	36	0	38
*StCPAI6*	Soltu.DM.07G016200.1	10.64	96aa	8.58	56.51	94.48	0.362	Extracellular	36	0	44
*StCPAI7*	Soltu.DM.07G016210.1	10.57	96aa	8.58	58.52	81.25	0.27	Extracellular	38	0	50
*StCPAI8*	Soltu.DM.07G016220.1	9.99	90aa	8.25	43.23	87.78	0.296	Extracellular	39	0	34

**Table 2 plants-14-02472-t002:** Primers for qRT-PCR.

Gene	Forward Primer Sequence (5′→3′)	Reverse Primer Sequence (5′→3′)	Amplicon Length
*qStCPAI2*	ATTGTGACTTGCGGGCATCGTTG	CGCAATGAGAAACCGCAGACCCA	100 bp
*qStCPAI3*	GGCCCAAGAAGATATACAACAAAT	CAAGTTTGTCTGAAATTCCAACAG	157 bp
*qStCPAI4*	GGCCCAAGAAGATCAAGATCCAAT	CAAGTTTGTCTGAAATTCCAACAG	105 bp
*qStCPAI5*	CAAGATGTGACGAAACTTTTTCAGGA	TTATTGAAAACTCTGGCGCCGC	147 bp
*qStCPAI7*	TCCCCCAGTGGTGGTACATACAG	CGCACGCACATAAACACACATAGA	94 bp
*qStCPAI8*	CCGCTTTAGTTAATATGCAAGTG	CAATCGGCGTTTGTGTTGCA	130 bp
*qACTIN11*	GGATCTTGCTGGTCGTGATTTAAC	CATAGGCAAGCTTTTCCTTCATGT	122 bp

## Data Availability

Data are contained within the article.

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
