# Peer review of "Identification and Characterization of the StCPAI Gene Family in Potato"

_plants, 2025, doi:10.3390/plants14162472_

Round 1

Reviewer 1 Report

Comments and Suggestions for Authors

General Comments

Reinforce Novelty: Clearly articulate the most significant novel findings presented in the paper in the conclusion. Impact Statement: Strengthen the concluding remarks on the practical implications or future impact of this research, especially for potato breeding or improving food security. By addressing these points, the authors can further enhance the clarity, impact, and reproducibility of their valuable research.

The paper presents a comprehensive bioinformatic and expression analysis of the StCPAI gene family in potato, which is a valuable contribution to understanding plant defense mechanisms and tuber development. The systematic approach to identifying and characterizing these genes is well-executed.

The use of various bioinformatics tools and experimental techniques (qRT-PCR) provides robust support for the findings. The discussion effectively links the findings to previous research in other Solanaceae species, strengthening the evolutionary and functional context of StCPAI genes.

Specific Comments and Suggestions

Introduction

The introduction is thorough but could be slightly more streamlined. For instance, the detailed description of general plant defense mechanisms (lines 36-45) could be condensed, as the focus quickly narrows to protease inhibitors.

While the importance of potato is well-stated (lines 28-35), explicitly reinforcing why studying CPAI in potato is particularly significant, beyond general plant defense, would strengthen the introduction. The current text mentions systematic characterization in potato hasn't occurred, which is a good starting point.

Results

Figure 1 - Scale Bar: Ensure the scale bar unit (Mb) is clearly visible and legible in the final published version of Figure 1.

Table 1 - Instability Index Interpretation: In line 97, the text states "collectively indicating that most StCPAIs are unstable alkaline hydrophobic proteins." While instability index values are provided, a brief explanation or a reference for the threshold value distinguishing stable from unstable proteins would be helpful for readers unfamiliar with this metric.

Figure 2 - Clarity: While it states "Tertiary structure prediction of StCPAI proteins," the figure itself isn't immediately clear what it's showing without context. Perhaps adding labels for individual proteins or a brief description within the figure legend could enhance clarity.

Phylogenetic Tree (Figure 3) - Legend Clarity: In the legend (lines 135-136), ensure that "Inner yellow stars, red squares, green circles, and purple checks represent St, Sl, Ca, and Sm CPAI proteins, respectively" is easily distinguishable visually in the figure (e.g., if colors are used, ensure they contrast well).

Collinearity Analysis (Figure 4) - Interpretive Text: Line 141-144 describes chromosomal locations. Ensure this text accurately reflects what is visually presented in Figure 4 and highlights the most significant collinear relationships. The current text mentions SlCPAI2 on Chr 3, then StCPAI2 on Chr 3, then SlCPAI3 on Chr 7, then StCPAI3 and StCPAI6 on Chr 7, and SlCPAI7 and SlCPAI10 on Chr 10. This seems to jump between species and chromosomes. Perhaps rephrase to better present the comparative locations for clarity.

Figure 5 - Sub-labels: Ensure that (a), (b), (c), and (d) are clearly labeled within Figure 5 for easy reference.

Figure 6 - Element Details: Consider if there's space to briefly explain what some of the more complex cis-acting elements represent (e.g., why is a "light response" element important here?) or categorize them more clearly in the legend.

Protein Interaction Network (Figure 7): The legend mentions "Grey lines indicate potential gene pairs with interaction relationships." If there are different types of lines or nodes, clarify their meaning (e.g., direct vs. indirect interaction, specific protein types) if possible.

Expression Patterns (Figures 8 & 9): Ensure that the axis labels and units are clear and large enough to read, especially for gene names and expression levels. For Figure 9, clearly indicate what "IAA treatment" represents (e.g., type of auxin).

Discussion

Synthesis of Findings: The discussion effectively summarizes the results. Consider beginning paragraphs with strong topic sentences that directly relate to the main findings of that section.

Deepen Functional Speculation: While the paper speculates on functions (e.g., StCPAI5 in stress/tuber sprouting, StCPAI8 in abiotic stress), expanding on the mechanistic implications or suggesting future experimental validation for these predicted interactions would strengthen the discussion. For example, how might the interaction with specific proteins lead to the observed phenotypes?

Role of StCPAI1: The specific high expression of StCPAI1 in aerial tubers (line 270) is very interesting. The discussion around it (lines 289-295) is good, but perhaps reiterate the potential significance for potato breeding or plant architecture in the context of this specific finding.

Comparison to SP6A: The comparison between StCPAI3/4 and SP6A is insightful (lines 273-279). Reconfirm that the distinction between "synthesized" and "accumulated" is clear to avoid ambiguity.

Future Directions: While the conclusion mentions future improvement targets, the discussion could benefit from a dedicated paragraph on specific future research directions, such as functional validation studies (e.g., gene knockout/overexpression), proteomic analysis, or detailed investigations into the regulatory networks involving StCPAI genes.

Materials and Methods

Access Dates: For all online databases and tools used (lines 327, 329, 332, 348, 349, 352, 354, 359, 371, 374, 375, 384, 390), ensure consistent and accurate "accessed on" dates.

Software Versions: Specify the exact versions of software used (e.g., TBtools, MEGA X, Cytoscape) to ensure reproducibility.

Primer Details (Table 2): For clarity, it would be beneficial to add the expected amplicon length for each primer pair, if space permits, or mention it in the text.

Reviewer 2 Report

Comments and Suggestions for Authors

The article is devoted to the identification and analysis of a small family of genes encoding Carboxypeptidase A inhibitor (CPAI) in potato plants. Eight genes have been identified. The structural features of these genes and the proteins encoding them have been studied, their localization on chromosomes has been shown, and a set of regulatory elements in gene promoters has been studied. The authors studied the regulation of gene expression under the influence of auxin using qRT-PCR. The article is very standard, based almost exclusively on bioinformatics analysis, despite this, it has some novelty with respect to the studied genes in potatoes. In terms of accumulating additional information about the studied genes, this work is useful.

Unfortunately, according to the reviewer, the manuscript has shortcomings that do not allow it to be recommended in this form for publication in a journal.

  1. The most serious comments on hormonal regulation of gene expression. The authors probably do not know the specifics of working with phytohormones. The concentration of auxin in any process should be selected by constructing a concentration curve. What was auxin dissolved in and was the same amount of solvent added to the control solution? The solvent can change the effect of the phytohormone. How and how many times were the plants treated. An incredibly high concentration of auxin, non-hormonal, was used. The hormone is not sodium chloride and in such a terribly high concentration it can be perceived by the plant not as a hormone, but as a stress factor. For this reason, all the results on auxin regulation of gene expression cannot be considered correct. This part of the work needs to be redone.
  2. The authors write that they failed to select successful primers for high-quality qRT-PCR for the StCPAI1 and StCPAI6 genes (Lines 211-212). This is quite strange in itself. But what is even more unexpected is that the unsuccessful primers are given in Table 2. In addition, the data on the expression of these genes are presented in Fig. 8 and 10, but they are not in Fig. 9. The primers are bad, and the results are given in the article.
  3. For some reason, very often, as in this article, the authors call organs tissues and talk about tissue-specific gene expression. As is known, organs consist of several or many tissues, so it is impossible to call organs tissues.
  4. The title of the article is questionable. Such little (and so far not high-quality) work has been done on hormonal regulation of gene expression that it makes no sense to include it in the title of the article.

Round 2

Reviewer 2 Report

Comments and Suggestions for Authors

Concerning this manuscript and the authors' responses, it seems that the authors are playing hide and seek with the reviewer. By the way, regarding the auxin concentrations, this is exactly the response I expected from the authors. This is the easiest way out of the difficult situation in which this work found itself. If the authors said so, we will assume that this is how it is. But I do not understand why the article specifies a stock solution of the hormone, which is stored in the freezer, but does not specify the concentration with which the plants were treated.

Regarding the bad primers, the authors' explanation cannot be accepted. What the authors write simply can never be. This is no longer about biology, but about chemistry. During RT-PCR, synthesized cDNA interacts under certain conditions with primers previously selected for this sequence. No matter what the plant is treated with, the primary sequence of the transcripts of the genes being studied will not change. This is especially impossible if the plant was treated with a phytohormone. So the authors' answer on this issue is not accepted.

There are two ways out of this situation.

  1. Exclude the results using these bad primers from the article.
  2. Refuse to publish this manuscript

Round 3

Reviewer 2 Report

Comments and Suggestions for Authors

The authors of the manuscript have corrected those parts of the article that I had comments on. And even unsuccessful primers and the results obtained with them are now presented more correctly. For this reason, I believe that the article can be accepted for publication.

P.S.

As for the authors' problem regarding the same primers working or not working with different types of plant material, this should never happen. In my opinion, there can be only two reasons: the authors processed the material in such a way that the expression of the gene being studied was suppressed (extremely unlikely) and the second - an error at one of the stages of the work - this is quite possible. Any reader who knows a little molecular biology will pay attention to this controversial moment in the work.